# An Influx of Non-Native Bird Species into the Natural Environment Owing to the Accidental Release of Pet Birds in Japan

**DOI:** 10.3390/ani14020221

**Published:** 2024-01-10

**Authors:** Sumiko Nishida, Wataru Kitamura

**Affiliations:** 1Environmental and Information Studies Division, Graduate School of Environmental and Information Studies, Tokyo City University, 3-3-1 Ushikubo-nishi, Tuzuki-ku, Yokohama 224-8551, Kanagawa, Japan; 2Department of Restoration Ecology and Built Environment, Faculty of Environmental Studies, Tokyo City University, 3-3-1 Ushikubo-nishi, Tuzuki-ku, Yokohama 224-8551, Kanagawa, Japan; kitamura@tcu.ac.jp

**Keywords:** biological invasions, bird introductions, exotic species, pet escape, cage-bird, naturalized parrot, monk parakeet, rose-ringed parakeet

## Abstract

**Simple Summary:**

The escape of pets plays a significant role in the establishment of invasive species. In Japan, where most pet birds in households are foreign-origin species, one study examined the number of these birds that escaped into the wild. Information on pet bird escapes was gathered from online lost-and-found websites for pets between 2018 and 2021. Escape incidents occur daily, with Budgerigars and Cockatiels being the most common escapees. Despite being popular pets worldwide, these birds rarely establish themselves outside their natural habitats. Monk Parakeets, which briefly bred in the wild in Japan in the 1980s but failed to establish themselves, and the Rose-ringed Parakeet, already established as an invasive species in the Kanto region, have also occasionally been reported as escapees. Considering their global invasive presence, it is crucial to be vigilant in preventing pet birds from escaping to protect local ecosystems. Regular escape from commonly maintained species highlights the need for responsible pet ownership and heightened awareness to curb potential environmental impacts.

**Abstract:**

The escape of pet birds into the wild raises concerns about the introduction of invasive avian species. This study investigated the impact of escaped pet birds on the introduction of non-native species in Japan. Data sourced from four lost-and-found pet websites between January 2018 and December 2021 revealed 12,125 recorded escapes exhibiting both daily occurrences and seasonal fluctuations. Statistical modeling identified the monthly average temperature (positively correlated) and maximum electricity demand (negatively correlated) as influential factors. Text analysis revealed “window” and “open” as frequently cited reasons for escapes. Budgerigars (*Melopsittacus undulatus*) and Cockatiels (*Nymphicus hollandicus*) accounted for 76% of the total escape, suggesting a low perceived risk of establishment in nonnative environments. Interestingly, two globally established invasive species, the Rose-ringed Parakeet (*Psittacula krameri*) and Monk Parakeet (*Myiopsitta monachus*), were among the escaped birds. While the Rose-ringed Parakeet is locally naturalized in Tokyo and its adjacent prefectures, the Monk Parakeet failed to establish itself in Japan. Despite the limited number of escaped Monk Parakeets, ongoing efforts are crucial for preventing the potential re-establishment of species with such capabilities.

## 1. Introduction

Pet bird keeping has been a human cultural activity practiced since ancient times in various parts of the world [1,2,3,4] and can be seen as representing a cultural aspect of the way humans interact with birds [5]. However, it has also become a source of various problems, including the decline of wild populations and the potential extinction of species due to overhunting [4,6], the introduction of invasive species by transporting non-native birds beyond their natural habitats [4,7,8,9], and ethical concerns related to the treatment of captive birds [10,11,12].

The introduction of non-native species through pet release and their potential impacts on natural ecosystems are serious concerns [7]. Japan imports the largest number of pet animals in the world [12,13] and it is expected that there will be considerable chances of pet escape. Although there have been several surveys on pet bird species sold in pet shops in Japan [14,15,16] or kept at home [17], pet bird escape in Japan has not been surveyed quantitatively because it is difficult to study and quantify the accidental escape of pet birds owing to their sporadic nature. However, this situation is concerning because most pet birds in Japan are non-native species, owing to restrictions on keeping Japanese wild birds [18], and the escape of pet birds from households can lead to the immediate introduction of non-native bird species into the wild. Thus, the detection and analysis of pet bird escape are urgently required.

One way to detect pet bird influxes is to obtain information provided by pet owners [19,20]. Studies using Internet resources have been gaining popularity, and datasets obtained from the Internet have been used in a variety of studies on pet-related invasive species [19,20,21,22]. When a pet bird escapes, its owner often posts information about the lost bird and asks for help. Such data on exotic pets available to observers via the Internet make it possible to detect bird escape events. Collecting and analyzing these data from the Internet provides useful information on the influx of non-native birds into natural ecosystems [19,20]. Rivas-Salvador et al. [20] investigated lost raptors in Spain and quantified the introduction rate of exotic raptors via falconry. Cassey and Vall-llosera [19] analyzed a dataset obtained from lost-and-found websites for pets in Australia and found that even in countries with strict regulations to control the inflow of non-native species from foreign countries, domestic pet birds were regularly escaping outdoors. Therefore, analyzing an Internet-sourced dataset on pet bird escape is important for understanding the potential introduction of non-native species into natural ecosystems, an aspect not previously explored in Japan.

The objective of our research is to retrospectively compute the escape of pet birds in Japan and determine the factors that influence their escape. Occasionally, exotic pet bird species are observed outdoors. However, it is not yet known if it is a rare event for potentially invasive bird species to enter the wild environment or, unfortunately, a daily occurrence that imposes significant propagule pressure on natural ecosystems. First, we counted the number of pet birds escaping using Internet resources. We also compared the list of escaped bird species with existing lists and recent articles on non-native bird species in Japan. Due to the escape of a substantial number of bird species listed in known non-native bird species compilations, we hypothesized that pet bird escapes may be influenced by factors facilitating access to the outdoors, such as doors and windows. We conducted a statistical analysis of the dataset of escaped birds to explore this hypothesis.

## 2. Materials and Methods

### 2.1. Study Area and Data Collection

The Japanese Archipelago comprises four large islands and more than 10,000 small islands, and due to its considerable length from north to south length, it encompasses a wide range of environments from cold temperate zones to the subtropics resulting in high biodiversity and the presence of several endemic species. However, the country has a high population density, with 120,000,000 people inhabiting a land area of 378,000 km^2^, but most of them are concentrated in coastal areas below 100 m above sea level, accounting for one-fourth of the country. Consequently, a series of large urban areas and highly diverse natural environments coexist.

There are naturalized bird species in Japan that are suspected to have originated from escaped or released pet birds, including Rose-ringed Parakeet (*Psittacula krameri*), Red-billed Leiothrix (*Leiothrix lutea*), Chinese Hwamei (*Garrulax canorus*), Moustached Laughingthrush (*Ianthocincla cineracea*), White-browed Laughingthrush (*Pterorhinus sannio*), Masked Laughingthrush (*Pterorhinus perspicillatus*), and Crested Myna (*Acridotheres cristatellus*) [23]. The Red-billed Leiothrix, Chinese Hwamei, Moustached Laughingthrush, White-browed Laughingthrush, and Masked Laughingthrush are designated as invasive alien species subject to legal control measures [24].

In Japan, birdkeeping is a popular practice [2], but capturing native wild birds for pet birdkeeping is restricted to protect the birds [18]. As a result, most pet birds in Japanese households are non-native bird species [14,17].

The dataset was obtained from lost-and-found websites for pets that escaped between January 2018 and December 2021. Several websites and social media broadcasts have lost pet information, but most of the information overlaps, and often the date, location, and species names are not well described on most websites and social networking service (SNS) accounts. Among the several websites that deal with lost pet information, four lost-and-found websites were selected because they were hosted by identifiable operators with stable management and had frequent posts (Appendix A). The selected websites covered all prefectures in Japan, and the posts were written in Japanese.

Data regarding species name, date of escape, location (prefecture, city, village), color variation (if applicable), behavioral traits (if applicable), cause of escape, and retrieval status, were collected from the websites.

### 2.2. Species Definition

To clarify the qualitative aspects of the accidental release of pet birds, we examined the types of birds that escaped most frequently by counting the total number of escapes for each bird species during the four years. The species name used in this study is based on the International Ornithological Congress (IOC) World Bird List v. 12.2 [25] except for the domestic species, including Ring-necked Dove (*Streptopelia roseogrisea*), Racing Pigeon and Fancy Pigeon (*Columba livia domestica*), Japanese Bantam (*Gallus gallus domesticus*), Bengalese Finch (*Lonchura striata domestica*), and Domestic Canary (*Serinus canaria domestica*). Because bird names and classifications used by the pet industry sometimes differ from the scientific names and classifications of birds, and the owners of escaped birds often use ambiguous words, it is sometimes impossible to identify the species’ names. In such cases, identification is limited to the genus level. All the conures of the genus Pyrrhura were combined as *Pyrrhura* spp. because they were often simply referred to as “Uroko Inko (Pyrrhura Parakeet)” on the websites; in those cases, the correct species names were not available. Similarly, *Agapornis fischeri*, *Agapornis personatus*, *Agapornis gapornis lilianae,* and *Agapornis nigrigenis* were combined as “*Agapornis* spp.” Barn owls were grouped as “*Tyto* spp.” Hybrid escapes were detected in Agapornis and Ara. The hybrid species Agapornis and Ara were denoted as *Agapornis* spp. and *Ara* spp., respectively.

There are several lists of non-native bird species in Japan. We have selected four of the lists and compared them to the detected bird escape. The four lists represented the *Check-list of Japanese Birds* by the Ornithological Society of Japan [23], the *List of Invasive Species of Japan* by the National Institute for Environmental Studies [26], the *Report of the Japan Breeding Bird Atlas* [27], and *the Handbook of Introduced Birds in Japan* [28], all of which were compiled after 2010, five years after the Invasive Alien Species Act (Act no. 78) [24] was enacted in 2005. Newly observed naturalized species obtained from recent studies [29,30] were also considered. Because each list was compiled from different perspectives, the definition of non-native bird species differed for each list. Non-native species at the subspecies level were included in *the Check-list of Japanese Birds* and *the Handbook of Introduced Birds in Japan* [23,28]. *The Handbook of Introduced Birds in Japan* also included reintroduced species, Crested Ibis (*Nipponia nippon*), Short-tailed Albatross (*Phoebastria albatrus*), Oriental Stork (*Ciconia boyciana*), free-range fowls, zoo-kept birds with the freedom to fly out to a certain extent, and Painted Stork (*Mycteria leucocephala*) [28]. Reintroduced species (Crested Ibis, Short-tailed Albatross, and Oriental Stork) were excluded from the comparison of the lists.

### 2.3. Extraction and Handling of Text Data

The owners of escaped birds often write about the cause of their escape in posts on lost-and-found websites. We obtained the cause of escape from the posts specifically in Tokyo, which is the most populated prefecture in Japan, where the largest number of birds escaped. This implies that the environment in Tokyo may be considered more uniform than in other prefectures. All the posts were written in Japanese. Texts describing the causes of escape were divided into morphemes using MeCab (version 0.996) [31] and KH Coder (version 3.0.0.0) [32]. Considering the grammatical structure of the Japanese language, only general verbs, common nouns, and sa-hen nouns were selected to obtain the frequency of their appearance in the text to identify common causes leading to bird escape.

### 2.4. Statistical Analysis

We investigated factors influencing the number of pet birds that escaped. The three most frequently appearing words in the causes of escape from posts on the lost-and-found websites for pets were “to escape” (nigeru), “window” (mado), and “to open” in transitive use (akeru) (Table 1).

We considered that the most frequently appearing word, “to escape”, did not play a significant role in determining the factors related to the escape because the word is often necessary to complete the sentence describing the cause of escape. The second most frequent word, “window”, and the third most frequent word, “to open” are considered important because when pet birds fly out, they have to fly through openings that connect the inside to the outside of rooms or cages. We focused on the factors that caused these openings. Assuming that windows were the most common openings that connect indoors and outdoors, regression analysis was performed using the average, maximum, and minimum temperatures, as well as precipitation and day length which are considered to be involved in opening and closing windows [33,34]. The maximum electricity demand was also considered because window opening and closing are related to the use of air conditioners [34], which account for the largest proportion of electricity consumed at home. Because winters in Tokyo are not extremely cold, the same air conditioners used for summer cooling are often used for winter heating. Regression analysis was performed using these factors as explanatory variables and the number of escapes as the response variable in R (version 4.3.0) [35] using the tidyverse package (version 2.0.0) [36]. Using the dredge function in R’s MuMIn package (version 1.47.5) [37], models with ∆AIC under 2.00 were selected.

The weather conditions dataset was obtained from the Japan Meteorological Agency (JMA) [38]. Day length was obtained from the Ephemeris Computation Office of the National Astronomical Observatory of Japan (NAOJ) [39]. The maximum electricity demand was obtained from the Organization for Cross-regional Coordination of Transmission Operators, Japan (OCCTO) [40].

We further investigated whether the second-highest-appearing word, “mado,” and the third-highest-appearing word, “to open” were correlated with the frequency of escapes. First, we examined the monthly changes in the appearance frequency of those words through regression analysis. The total month was used as an explanatory variable, and the monthly appearance frequency of each word (or the number of monthly appearances of each word) was considered as the response variable. The monthly appearance frequency of a word represented the number of occurrences of each word divided by the total number of occurrences in the corresponding part of speech. Second, we conducted a regression analysis with temperature as an explanatory variable and the number of monthly appearances of each word as a response variable.

## 3. Results

A total of 12,126 escapes were detected over the four years from 2018 to 2021 (Appendix A). Escapes occurred in all prefectures in Japan (Figure 1). The number of escapes was high in prefectures with large cities and populations, and the prefecture where the most frequent escapes occurred was Tokyo. The escape rate was high in summer and low in winter (Figure 2), with no instances of escape on single days.

### 3.1. Species Variation

More than 60 escape species were identified during the study period (Appendix A). Among the escaped species, 91% were Psittaciformes.

Budgerigar (*Melopsittacus undulatus*) and Cockatiel (*Nymphicus hollandicus*) were the two species with the highest escape frequency, accounting for 76% of the total number of escaped birds (Figure 3). These were followed by Java Sparrow (*Padda oryzivora*, (7.7%)), Rosy-faced Lovebird (*Agapornis roseicollis*, (6.8%)), Pyrrhura parakeet (*Pyrrhura* spp., (1.9%)), Lilian’s Lovebird (*Agapornis lilianae*, (1.8%)), Pacific Parrotlet (*Forpus coelestis*, (1.1%)), Barred Parakeet (*Bolborhynchus lineola*, (0.8%)), Monk Parakeet (*Myiopsitta monachus*, (0.6%)), and Zebra Finch (*Taeniopygia guttata*, (0.5%)). (Figure 3). Over ten species accounted for 98% of the total escape. In the present study, escaped Parrots of the genus Psittacula were detected and categorized as non-native Japanese species. The native Japanese species reported to have been lost include Accipitriformes, Falconiformes, and Galliformes. These include the Northern Goshawk (*Accipiter gentilis*) (Accipitriformes), Peregrine Falcon (*Falco peregrinus*) (Falconiformes), and Japanese Quail (*Coturnix japonica*) (Galliformes).

### 3.2. Factors Influencing Pet Bird Escape

Model selection using generalized linear models showed that the best-fit model was the one with two explanatory variables, including maximum demand and average temperature. Maximum demand had a negative relationship with the number of escapes (−0.4499), while average temperature had a positive relationship (0.99505). Also, the models with maximum electricity demand as explanatory value and the models with average temperature as explanatory value were most frequently adopted in the selected models (∆AIC < 2.00) (Table 2).

### 3.3. Causes of Escape

A total of 10,395 posts on escapes contained text describing the cause of the escape. In the 20 most frequently appeared words, the words related to the openings that connect the inside to the outside, including “window” (mado), “entrance” (genkan), “balcony” (beranda), “door” (doa), and another Japanese word for “door” (tobira), accounted for 27% of the total word appearances (Table 1). The words related to human activity, including “open” (akeru), “releasing a bird in a room” (ho-cho), “clean up” (so-ji), and “carelessness” (fuchuui) also appeared frequently (Table 1).

The appearance frequency of “window” and “to open” did not show temporal fluctuations (Figure 4). Regression analysis showed that there was no correlation between temperature and appearance frequency of either “window” (coefficient = 0.032, *p* = 0.698) or “to open” (coefficient = −0.159, *p* = 0.063).

## 4. Discussion

### 4.1. Factors Involved in the Number of Escapes

The number of escapes fluctuated throughout the four years, and these fluctuations seemed to be related to seasonal changes. The model selection results showed that the maximum electricity demand and average temperature were most closely related to the number of escapes, with the former exhibiting a negative relationship and the latter exhibiting a positive relationship. An increase or decrease in the maximum electricity demand corresponds to an increase or decrease in the number of air conditioners used at home [36]. The number of air conditioners used also affects the opening and closing of windows [33,34]; people are more likely to close their windows when using an air conditioner [34]. This means that as the number of air conditioners used increases the number of open windows, and thus the number of routes for runaways decrease.

Temperature changes with the season, making it the major factor in seasonal variation. Additionally, day length varies significantly depending on the season. Temperature and day length photoperiod are known to influence human and bird activity [41,42,43]. However, day length has rarely been selected as the best model. Temperature may be indirectly related to escape numbers, for example, through the width of a window opening [34].

An analysis of texts on the causes of escape also suggests that opening windows is an important reason (Table 1). However, the appearance frequencies of words closely related to openings did not show seasonal fluctuations and remained stable throughout the study period. In addition, the appearance frequency of the words did not correlate with temperature, which changed seasonally. This implies that, although the number of escapes and the posting of the cause of escape fluctuated, the proportion of bird escapes involving windows remained constant for all reasons throughout the year. Window opening is related to temperature [33,34]. However, words related to window opening did not seem to fluctuate with temperature. This suggests that factors other than window opening may have caused the fluctuations in the number of escapes. We extracted text data on the causes of escapes in Tokyo and there may be some bias. The architectural features of homes can vary between the metropolitan area of Tokyo and non-urban areas in rural Japan. Specifically, traditional houses in non-urban areas may have more and wider openings [34] and potentially have more opportunities for windows and doors to be left open on a daily basis. Additionally, in regions in the northern part of the Japanese archipelago, air conditioners are not commonly used for heating during winter, so electricity consumption in winter does not necessarily reflect the use of air conditioners.

### 4.2. Species

In this survey, 91% of the domestic birds confirmed to have escaped were of the order Psittacidae. This suggests that parrots may have escaped more than other species. However, it is possible that the data obtained from the lost-and-found websites for pets were biased toward parrots. Pet parrots are often taken out of their cages to play with their owners, to receive training, or to roam around freely [44,45] in environments where there are more chances of escaping outside, including living rooms with doors and windows that owners may carelessly open.

While the number of escapes was biased towards parrots, bird species of other orders could be biased. Raptors were detected in small numbers during the study period. Pet raptors have a higher chance of escaping because they need to fly freely outside for exercise, training, hunting, and other activities [46]. Furthermore, individuals with falcons as pets may form their own group and share information within this group rather than on lost-and-found websites for pets [20]. Additionally, in recent years, it has become common for raptors to be equipped with radio transmitters during their free flight [20], and escape is more likely to be a matter of immediate tracking using tracking systems and automobiles rather than posting on lost-and-found websites. The escape of Finches other than Java Sparrow and Zebra Finch was also detected in small numbers, although finches are popular and often kept as pet species in Japan [16]. How most finch species are maintained is often different from how owners maintain parrot species. They were not trained to fly in the room and return to their cage, except for some species, such as the Java Sparrow and Zebra Finch. They were more likely to stay in their cages, and if they escaped, it was extremely difficult to retrieve the birds. This may have caused the owner to refrain from posting an escape on lost-and-found websites.

A total of 6469 Budgerigars and 2750 Cockatiels escaped during the four-year period, making them the bird species with the highest escape frequency. Both are Australian Psittaciformes species. The Budgerigars and Cockatiels are nomadic birds, which are unlikely to become established as invasive species [47,48,49,50]. Cockatiels are not on Japan’s list of invasive species. Budgerigar is listed as an invasive species among the four Japanese invasive species (Appendix A). One of the lists, *Report of the Breeding Bird Atlas of Japan*, shows that although the breeding of budgerigars was confirmed in Japan in the 1970s, breeding or inhabitation was no longer observed during the most recent survey period (2016–2021) [27]. Budgerigar is the most commonly kept pet bird worldwide [49,50], and there are many escapes around the world [48], but there have been very few confirmed cases of these birds establishing themselves outside their natural habitat [49]. The only exception is the U.S. state of Florida, where escaped or intentionally introduced Budgerigars have established wild populations and were bred in the past [50,51]. However, these populations, which numbered in thousands in the 1970s and the 1980s, rapidly reduced in number from the late 1980s and are thought to have disappeared after being last sighted in 2014 [52]. Additionally, these two species are captive-bred and may face difficulties in establishing themselves in comparison to the wild-caught birds [4]. Therefore, these two bird species, which accounted for more than three-quarters of the escaped birds, are considered to have a low impact as sources of invasive species.

Java Sparrows (*Lonchura oryzivora*) were the third-highest among the escaped species (Figure 3). This species has been established as an invasive species in some areas of Southeast Asia and the Hawaiian Islands of the Pacific Ocean [53]. Previously, free-living Java sparrows have been observed. They were observed to form flocks outdoors in the 1920s when a big pet boom suddenly collapsed due to economic depression and a large number of domestic birds were carelessly released [54]. Later, outdoor breeding was observed in the 1970s but was no longer observed in the 1990s or after [27]. This was considered a failure to establish themselves by the bird species [26]. In Japan, even if escaped Java Sparrow individuals temporarily live outdoors, it may be difficult to establish a fully stable population. However, because these birds have been observed outdoors for more than 100 years and have become invasive birds in some regions of Asia and the Pacific Islands [53], the number of escapes should be kept as low as possible.

A total of 71 Monk Parakeets, an invasive species worldwide [55,56], were reported to have escaped within the four-year study period. In the 1980s, Monk Parakeets were observed for several sequential years [57]; however, there are currently no reports on established populations [27]. However, since it was found that Monk Parakeets keep escaping, the owners need to be careful and not allow their escape to the outside environment. Rose-ringed Parakeets are also species that have expanded their distribution worldwide [56,58], and in this study, 14 escapes of these species were reported. Naturalized Rose-ringed Parakeets were first observed in the late 1960s [59,60] and now inhabit the metropolitan area of the Kanto Plain, consisting of Tokyo and neighboring prefectures [Matsunaga]. The population size of Rose-ringed Parakeets was estimated to be approximately 2000 in 2018 [61].

Although Red-billed Leiothrix, Chinese Hwamei, Moustached Laughingthrush, White-browed Laughingthrush, and Masked Laughing thrush are considered to be pet-origin invasive species in Japan [26], none of their escapes were detected in this study. They are classified as important invasive species in Japan by the Ministry of Environment [24] and keeping them as pets is prohibited legally [24]. Thus, few owners keep these species as pets. Even if there were a few birds, they would not post a message on the websites stating that their illegally kept bird had escaped.

## 5. Conclusions

When considering the conservation of the natural environment, pet birds in cages tend to be ignored unless the study directly deals with the impact of the pet trade on birds in their native range. One reason is that pet birds are assumed to be under the complete control of their owners and stay in an environment isolated from the outdoors. Thus, pet bird keeping is more likely to be seen as a human activity than things related to a part of nature. However, it has been discovered that pet birds escaping their cages and venturing into the wild environment is an everyday occurrence. Budgerigar and Cockatiel species, which escaped in large quantities in this study, can be considered to have a low risk of establishing themselves as naturalized parrots in Japan due to their traits [47]. However, the Monk Parakeet, which has not been established in Japan but naturalized in various places in the world, as well as the Rose-ringed Parakeet, which has already been established in Japan, have been detected to have escaped. Although only a small number of Monk Parakeets and Rose-ringed Parakeets were detected, the constant escape of species that have been proven to become invasive could be a source of another invasive species in Japan or another population of invasive species in different areas of Japan. Therefore, if they live as pets, it is extremely important to prevent their escape.

Recently, information about escaped pet birds is being shared not only on lost and found websites for pets but also increasingly on SNS. Thoroughly examining these online pieces of information can increase the chances of an escaped bird returning to its owner. Furthermore, it may contribute to the development of an early warning system for the invasion of non-native species for natural environment managers.

## Figures and Tables

**Figure 1 animals-14-00221-f001:**
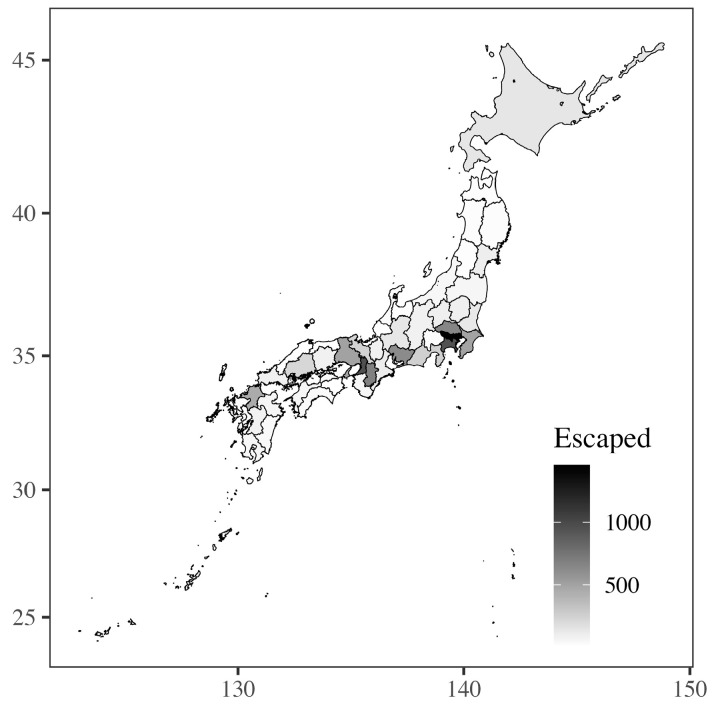
Geographic variation in the frequency of pet bird escape in Japan.

**Figure 2 animals-14-00221-f002:**
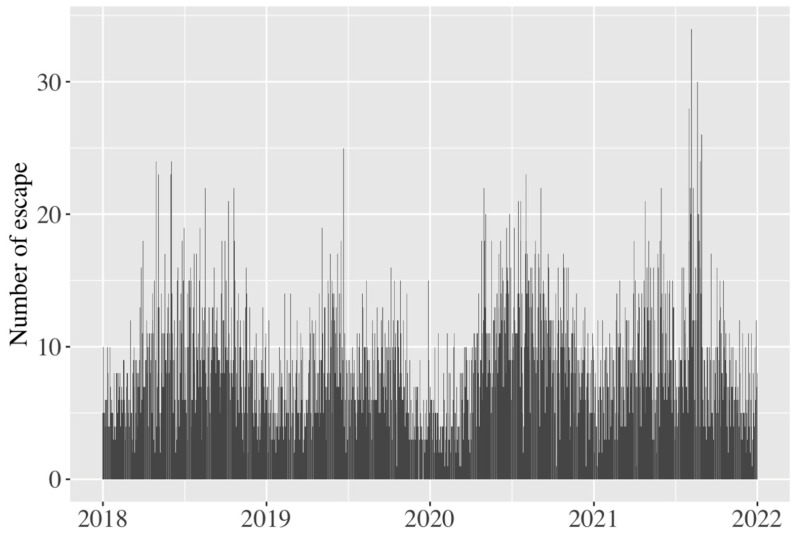
Temporal variations in number of escapes per day.

**Figure 3 animals-14-00221-f003:**
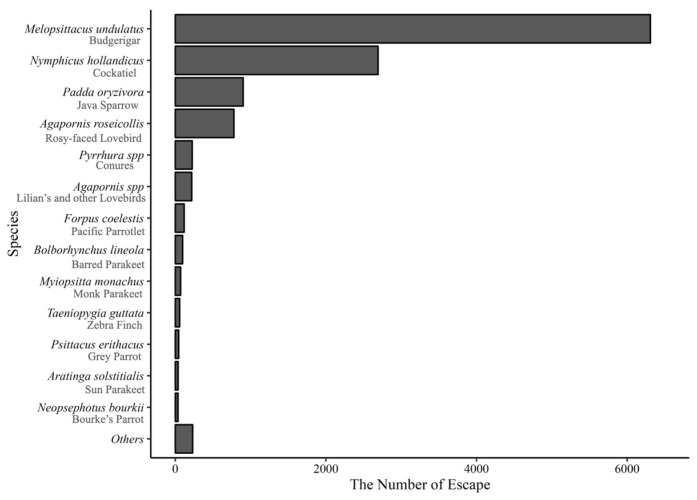
Species of pet birds most frequently reported as escaped.

**Figure 4 animals-14-00221-f004:**
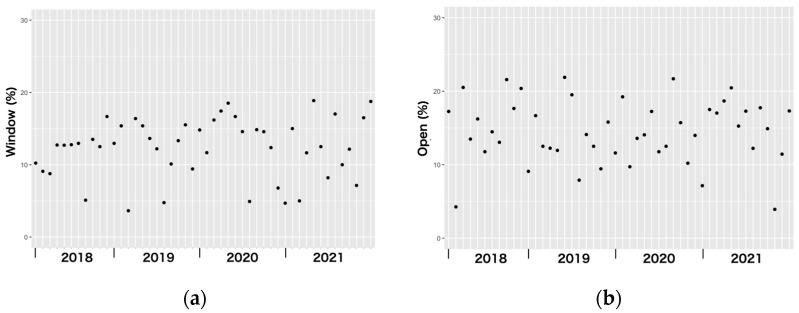
The appearance frequency of the most frequently appearing words in the texts posted by owners related to the cause of escape; (**a**) “window (mado)”; and (**b**) “to open (akeru)”.

**Table 1 animals-14-00221-t001:** The most frequently used words (nouns and verbs) were translated from Japanese to English. The “noun sa” is a type of noun (sa-hen noun) that can be transformed into a verb. The top 20 words are depicted.

Japanese	English	Type	Number
nigeru	escape	verb	2738
mado	window	noun	2689
akeru	open (transitive)	verb	2679
tobu	fly	verb	2570
deru	go out	verb	2491
kago	cage	noun	2041
genkan	entrance	noun	2040
beranda	balcony	noun	1949
okonau	do	verb	1811
soto	outside	noun	1721
tobidasu	jump out of	verb	1437
ho-cho	releasing a bird in a room	noun sa	1338
aku	open (intransitive)	verb	1263
doa	door	noun	1202
kazoku	family	noun	865
kata	shoulder	noun	814
soji	clean up	noun sa	746
tobira	door	noun	678
fuchu-i	carelessness	noun	619

**Table 2 animals-14-00221-t002:** Model selection using generalized linear models. Models with a delta Akaike’s Information Criterion (AIC) of less than 2.0 have been selected.

	Maximum Electricity Demand	Precipitation	Temperature Average	Temperature Maximum	Temperature Minimum	AIC	∆AIC
1	−0.4499		0.9505			366.93	0.00
2			0.8581			368.13	1.20
3	−0.4381			1.0012		368.52	1.60
4	−0.4234	0.0100	0.8782			368.54	1.62
5	−0.4195		1.5293		−0.5697	368.59	1.66
6	−0.4780				0.8731	368.81	1.89
7	−0.4485		1.1514	−0.2266		368.87	1.95
NULL	–	–	–	–	–	382.40	15.48

## Data Availability

Weather data can be uploaded from [38]. Electricity data can be obtained from [40].

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
