# Peer review of "An Influx of Non-Native Bird Species into the Natural Environment Owing to the Accidental Release of Pet Birds in Japan"

_animals, 2024, doi:10.3390/ani14020221_

Round 1

Reviewer 1 Report

Comments and Suggestions for Authors

I would like to make several observations that I think would improve the work. First, the list of species includes species reintroduced into the wild, which should not be considered exotic, as they are reintroduced because they were or are considered local (Ciconia boyciana and  Nipponia nippon). Second, I believe that the conclusions can indicate the value for natural environment managers that web reviews can have to establish an early warning system that would quickly detect the presence of the most potentially invasive species such as the Kramer’s parrot and proceed to try to control them as soon as possible. Third, youy must include right reference in line 180. Fourth,  do believe that the bibliographic references are not all appropriate to the usual standards (for example, see references numbers 2 and 4). I think it would be good you could review the entire list to make it homogeneous

Reviewer 2 Report

Comments and Suggestions for Authors

The manuscript addresses the issue of accidental avian introductions due to pet escapes in Japan, and uses a novel method to obtain data on birds released from lost-and-found websites on the internet. The authors are commended for addressing an issue that few have attempted to quantify. The analysis of the data is adequate and shows a clear trend of increased numbers of birds escaping during the summer. The potential impact of the species being released on the local environment is addressed in the discussion, as are the potential reasons why passerine escapes are probably less likely to be reported by owners. 

I have some minor comments on points that could be clarified in the manuscript below: 

Line 155–157: Does Tokyo differ from the rest of Japan in important ways for this analysis? It is important to acknowledge possible sources of bias in the discussion when considering how generalisable your findings are to the whole of Japan.

 Line 254–263: The discussion about air conditioners will be confusing to many non-Japanese readers I think. I guess that air conditioners are also used to heat homes in the winter? If that is true, then it is important to consider that most people relate air conditioners to hot weather and cooling a home, and would therefore expect their use only in the summer (and therefore closed windows and lower risk of bird escape). It would be clearer to change the use of air conditioner for heating or something similar.

 Line 178–180 would be a good place to describe the use of air conditioners in Japan for an international readership.

 Line 17 – “which were briefly bred in the wild”. This is a minor English error but it changes the meaning to imply the birds were being bred by humans in the wild. Remove the “were”.

Comments on the Quality of English Language

The English is easy to follow. However, there are inconsistencies in capitalisation (e.g., Internet and internet, species names), and the manuscript would benefit from minor editing by a professional who can clear up the few errors that appear. 
